# Arbidol inhibits human esophageal squamous cell carcinoma growth in vitro and in vivo through suppressing ataxia telangiectasia and Rad3-related protein kinase

Ning Yang[1,2†], Xuebo Lu[1,2†], Yanan Jiang[1,2,3,4†], Lili Zhao[1], Donghao Wang[1,2], Yaxing Wei[1,2], Yin Yu[1,2], Myoung Ok Kim[5], Kyle Vaughn Laster[2], Xin Li[1,3], Baoyin Yuan[1,3], Zigang Dong[1,2,6*], Kangdong Liu[1,2,3,4,6*]

[1]Pathophysiology Department, The School of Basic Medical Sciences, Zhengzhou University, Zhengzhou, China; [2]China-US Hormel Cancer Institute, Zhengzhou, China; [3]Collaborative Innovation Center of Henan Province for Cancer Chemoprevention, Zhengzhou University, Zhengzhou, China; [4]State Key Laboratory of Esophageal Cancer Prevention and Treatment, Zhengzhou, China; [5]Department of Animal Science and Biotechnology, Kyungpook National University, Sangju, Republic of Korea; [6]Cancer Chemoprevention International Collaboration Laboratory, Zhengzhou, China

*For correspondence:
dongzg@zzu.edu.cn (ZD);
kdliu@zzu.edu.cn (KL)

+Ning Yang, Xuebo Lu and Yanan Jiang contributed equally to this work.

Competing interest: The authors declare that no competing interests exist.

**Abstract** Human esophageal cancer has a global impact on human health due to its high incidence and mortality. Therefore, there is an urgent need to develop new drugs to treat or prevent the prominent pathological subtype of esophageal cancer, esophageal squamous cell carcinoma (ESCC). Based upon the screening of drugs approved by the Food and Drug Administration, we discovered that Arbidol could effectively inhibit the proliferation of human ESCC in vitro. Next, we conducted a series of cell-based assays and found that Arbidol treatment inhibited the proliferation and colony formation ability of ESCC cells and promoted G1-phase cell cycle arrest. Phosphoproteomics experiments, in vitro kinase assays and pull-down assays were subsequently performed in order to identify the underlying growth inhibitory mechanism. We verified that Arbidol is a potential ataxia telangiectasia and Rad3-related (ATR) inhibitor via binding to ATR kinase to reduce the phosphorylation and activation of minichromosome maintenance protein 2 at Ser108. Finally, we demonstrated Arbidol had the inhibitory effect of ESCC in vivo by a patient-derived xenograft model. All together, Arbidol inhibits the proliferation of ESCC in vitro and in vivo through the DNA replication pathway and is associated with the cell cycle.

## Editor's evaluation

This manuscript will be of interest to a broad audience of cancer biologists, especially those interested in esophageal cancer or treatment strategies involving ATR inhibition. It provides novel information about how FDA-approved antiretroviral compound Arbidol is a potential ATR inhibitor, which is of interest in the treatment of multiple tumor types. The key claims of the manuscript are supported by in silico, in vitro, and in vivo data.

## Introduction

Human esophageal cancer is a malignant neoplasm with a high incidence worldwide, ranking eighth in global incidence and sixth in overall mortality (*Bray et al., 2018*; *Zhang et al., 2021*). It has two pathological types: esophageal squamous cell carcinoma (ESCC) and esophageal adenocarcinoma (*Napier et al., 2014*). ESCC is the main histological type and accounts for approximately 90% of esophageal cancers in China (*Ke et al., 2019*). Current treatment options for patients with ESCC are surgery, chemotherapy, and radiotherapy (*Ha et al., 2020*). However, the 5-year survival rate remains less than 20% (*Pennathur et al., 2013*). Therefore, there is an urgent need for effective therapeutic or chemopreventive agents to improve the survival rate of patients with ESCC.

Chemoprevention is a promising strategy to prevent the recurrence of cancer (*Liu et al., 2008*). In recent years, studies have found that certain non-steroidal anti-inflammatory drugs can be used for this purpose (*Bacchi et al., 2012*). For example, it was reported that aspirin was used for chemoprophylaxis of colorectal, prostate, and other cancers (*Gronich and Rennert, 2013*). Aside from non-steroidal anti-inflammatory drugs, other repurposed drugs also exhibit antitumor effects. Metformin, a hypoglycemic drug has been shown to have a preventive effect on lung cancer, gastric cancer, pancreatic cancer, breast cancer, ovarian cancer, colorectal cancer, liver cancer, and other cancers (*Duncan and Schmidt, 2009*; *Morales and Morris, 2015*). Therefore, it is likely that repurposing pre-existing Food and Drug Administration (FDA)-approved drugs could also be a promising strategy to prevent ESCC recurrence.

We screened a series of FDA-approved drugs and found that Arbidol has a significant inhibitory effect on the proliferation of ESCC cells. Arbidol is a broad-spectrum antiviral compound which is conventionally used for the prevention and treatment of influenza (*Blaising et al., 2014*). Its antiviral activity against a variety of DNA and RNA viruses is mainly based on disrupting key steps during the virus-cell interaction (*Zhong et al., 2009*). However, its role in cancer prevention or treatment has not been reported. Here, we identified that Arbidol is an inhibitor of the ataxia telangiectasia and Rad3-related (ATR) protein kinase.

ATR belongs to the phosphoinositide (PI) 3-kinase-related protein kinase (PI3KKs) family and one of its key functions is to regulate DNA replication (*Martínez et al., 2014*; *Qiu et al., 2018*). ATR can phosphorylate the key replication protein minichromosome maintenance protein 2 (MCM2) on serine 108 in response to various forms of DNA damage and stalling of replication forks (*Hae et al., 2004*; *Lecona and Fernandez-Capetillo, 2018*). In recent years, ATR inhibitors have been used in many cancer investigations (*Yuan et al., 2019*), and their roles in cancer treatment have gradually been realized.

We performed mass spectrometry (MS) analysis of lysates derived from Arbidol-treated cells to investigate the proteomic and phosphoproteomic response upon treatment. Additionally, we performed in silico molecular docking on a panel of kinases to predict potential binding partners and clarify the drug's inhibitory mechanism. Our results showed that Arbidol inhibited ESCC growth via modulating the MCM2-ATR axis.

## Results

### Arbidol inhibits ESCC cell proliferation

To identify a drug able to inhibit ESCC cell proliferation, we conducted toxicity experiments using FDA-approved drug libraries and discovered that Arbidol (*Figure 1a*) effectively suppresses ESCC cell growth (*Figure 1b*). The IC50 values of Arbidol in KYSE150, KYSE450, and shantou human embryonic esophageal (SHEE) cells were 57.54, 54.72, and 87.63 μM at 24 hr and 26.37, 20.75 and 62.52 μM at 48 hr, respectively (*Figure 1c*). We found that Arbidol had a higher IC50 value in the SHEE immortalized esophageal epithelium cell line than in ESCC cells, which indicated that Arbidol had stronger toxicity in ESCC cells. To further evaluate the effect of Arbidol on ESCC growth, we treated KYSE150 and KYSE450 cells with different concentrations (0, 2.5, 5, 10, and 20 μM) of Arbidol (*Figure 1d*). The results indicated that Arbidol inhibited the proliferation of KYSE150 and KYSE450 cells in a dose-dependent manner. It is worth mentioning that we also performed toxicity and proliferation experiments on the SHEE human immortalized esophageal epithelial cell line and observed that the growth inhibitory effect was less pronounced. We next assessed whether treatment with Arbidol could impact anchorage-independent cell growth using the soft agar assay. The result showed that

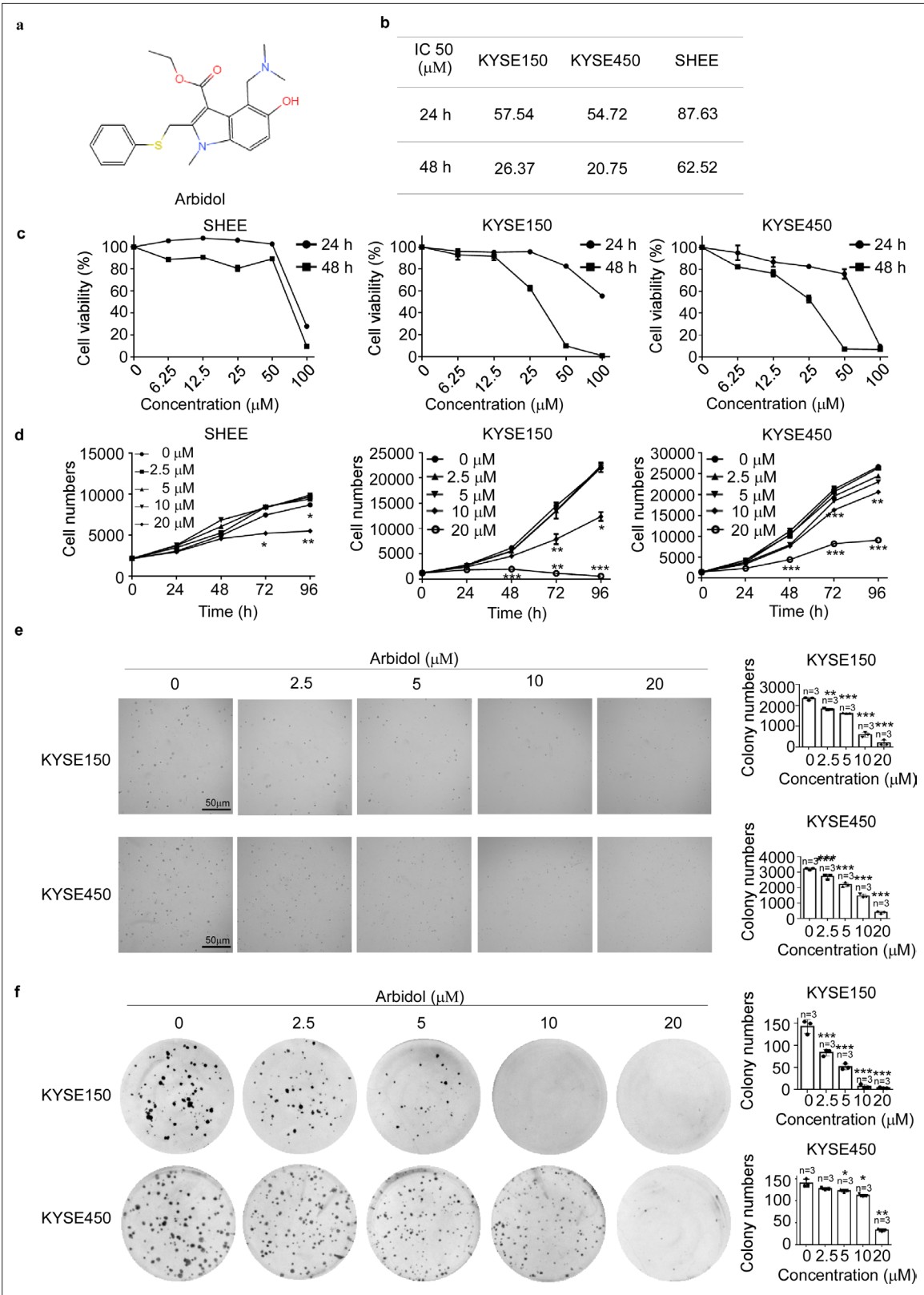

**Figure 1.** Arbidol inhibits ESCC cell proliferation. (**a**) The structure of Arbidol. (**b**) KYSE150, KYSE450, and shantou human embryonic esophageal (SHEE) cell viability was detected by the IN Cell Analyzer. KYSE150 and KYSE450 cells were treated with different concentrations of Arbidol (0, 6.25, 12.5, 25, 50, and 100 µM) for 24 hr and 48 hr and evaluated by 4',6-diamidino-2-phenylindole. IC50 values were calculated by SPSS 24.0 software (IBM, USA). (**c**) KYSE150, KYSE450, and SHEE cells were treated with Arbidol (0, 6.25, 12.5, 25, 50, and 100 µM) at different concentrations as indicated for 24 hr and

*Figure 1 continued on next page*

*Figure 1 continued*

48 hr. Cell viability was then determined by IN Cell Analyzer. (**d**) The effect of different concentrations of Arbidol (0, 2.5, 5, 10, and 20 µM) on SHEE, KYSE150, and KYSE450 cells for 0, 24, 48, 72, and 96 hr. Data were analyzed by homogeneity of variance and one-way analysis of variance (ANOVA). Asterisks indicate a significant decrease *p<0.05, **p<0.01, ***p<0.001, respectively. (**e**) Arbidol effectively inhibits anchorage-independent cell growth. Colony numbers were counted using an IN Cell Analyzer. Data were analyzed by homogeneity of variance and one-way ANOVA. Asterisks indicate a significant difference, *p<0.05, ** p<0.01, and ***p<0.001, n=3, compared with the control group. (**f**) Arbidol effectively inhibits colony formation in esophageal squamous cell carcinoma (ESCC) cell lines. KYSE150 (200/well) and KYSE450 (400/well) cells were seeded in 6-well plates and treated with different concentrations of Arbidol (0, 2.5, 5, 10, and 20 µM). The numbers of cell colonies were counted after 10–14 days. Data were analyzed by homogeneity of variance and one-way ANOVA. Asterisks indicate a significant difference, *p<0.05, ** p<0.01, and ***p<0.001, n=3, compared with the control group.

Arbidol inhibited anchorage-independent growth of KYSE150 and KYSE450 cells in a dose-dependent manner (*Figure 1e*). Finally, we used the colony-formation assay to determine whether Arbidol inhibited the proliferation of ESCC cell lines (*Figure 1f*). Similarly, the results indicated that Arbidol inhibited colony-formation in a dose-dependent manner. Collectively, these results suggested that Arbidol had a significant inhibitory effect on ESCC in vitro.

## Phosphoproteomics reveals that Arbidol inhibits tumors through the MCM-ATR signaling pathway

To determine the specific inhibitory mechanism of Arbidol in ESCC, KYSE150 cells were treated with 20 µM Arbidol or dimethyl sulfoxide as control for 24 hr. Subsequently, MS was performed (*Figure 2a*). The MS proteomics data have been deposited to the ProteomeXchange Consortium (http://proteomecentral.proteomexchange.org) via the iProX partner repository with the dataset identifier PXD034944. We identified 241 downregulated proteins and 340 downregulated sites. Next, we performed KEGG enrichment analysis using the quantified phosphate sites to query significantly downregulated pathways (*Figure 2b and c*). The DNA replication was identified as the key pathway affected. A heat map detailing phosphorylation site enrichment and KEGG pathway enrichment showed that the S108 site of MCM2 in the DNA replication pathway was significantly downregulated after Arbidol treatment (*Figure 2d and e*). The molecular profile of MCM2 S108 obtained using MS was displayed in *Figure 2f*. The results of the in silico analysis were verified by western blot (*Figure 2g*). The results of western blot confirmed the phosphoproteomics data and showed that MCM2 S108 phosphorylation was strongly inhibited after Arbidol treatment. To further investigate the specific targets of Arbidol, we first searched for proteins upstream of p-MCM2 Ser108 and found that Cdc7 and ATR can phosphorylate MCM2 (Ser108) (*Charych et al., 2008*; *Cheng et al., 2017*; *Huang et al., 2019*). However, pull-down results indicated that Cdc7 could not directly bind to Arbidol (*Figure 2—figure supplement 1*). Next, we conducted an in-depth analysis of the recent quantitative human interaction group (pull-down MS), and utilized the kinase prediction tool (from phosphoproteomics data). We also used the chemical structure of Arbidol to predict targets using the SwissTargetPrediction. The results suggested that ATR kinase was a potential target of Arbidol.

## Arbidol targets ATR to affect DNA replication pathway in ESCC

We conducted a computational docking study using the molecular structures of Arbidol and ATR to determine optimal binding orientation. The results suggested that Arbidol could bind to ATR at the ASN 2346 and HIS 2361 residues (*Figure 3a*). Based on the sample information from pull-down MS, we found that 906 candidate targets were differentially represented between the control and treatment groups. Next, we verified that ATR, one of the targets, could bind to Arbidol through a pull-down assay in vitro and ex vivo (*Figure 3b-d*). To validate the results of the molecular docking, as well as to assess the contribution of ASN 2346 and HIS 2361 to the binding between Arbidol and ATR, we mutated these two residues and performed a pull-down assay. The results showed that the binding between Arbidol and ATR was less efficient after ATR was mutated (*Figure 3e*). In addition, the results of an in vitro kinase assay indicated that Arbidol treatment reduced MCM2 phosphorylation levels (*Figure 3f*, *Figure 3—figure supplement 1*). The ATP competition assay further illustrated that Arbidol functions as an ATP-competitive inhibitor of ATR (*Figure 3g*). These results indicated that ATR was involved in the phosphorylation of MCM2, and that ATR inhibition induced by Arbidol treatment may affect the phosphorylation of MCM2.

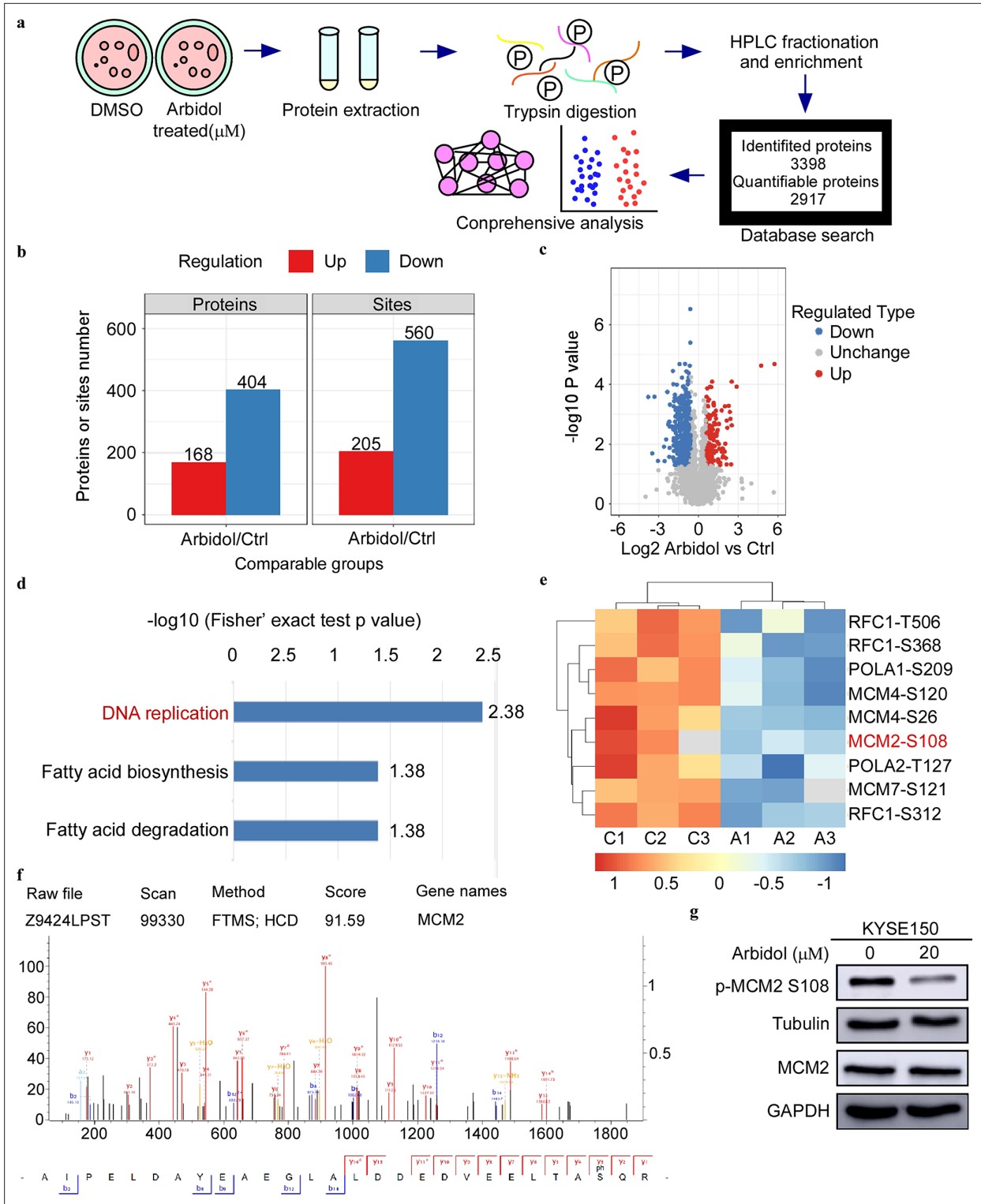

**Figure 2.** Phosphoproteomics reveals that Arbidol inhibits tumors through minichromosome maintenance-ataxia telangiectasia and Rad3-related (MCM-ATR) signal pathway. (**a**) The mass spectrometry workflow of phosphorylated proteomics analysis of KYSE150 cells after 24 hr treatment with Arbidol. (**b**) Histogram shows the regulation of phosphorylation sites compared to the control group. (**c**) The volcano plot shows that 375 phosphorylation sites changed significantly. (**d**) Kyoto Encyclopedia of Genes and Genome (KEGG) analysis indicated that three downregulated pathways were enriched. (**e**) Phosphorylation proteomics analysis of esophageal squamous cell carcinoma (ESCC) cell after 20 µM Arbidol treatment. The different phosphorylation sites were plotted as heat maps. (**f**) Distribution of peptide mass error and length of protein peptides identified in this study. (**g**) Western blot analysis shows that the phosphorylation of MCM2 S108 is markedly downregulated after 24 hr of Arbidol treatment.

*Figure 2 continued on next page*

Figure 2 continued

The online version of this article includes the following source data and figure supplement(s) for figure 2:

**Source data 1.** The original data of western blot in *Figure 2*.

**Figure supplement 1.** Arbidol did not bind directly to CDC7.

**Figure supplement 1—source data 1.** Arbidol did not bind directly to CDC7.

## Arbidol inhibits the proliferation of ESCC cells and arrests the cell cycle at G1-phase

To elucidate the functional pathways of ATR, a Gene set enrichment analysis (GSEA) was performed using our phosphoproteomics and proteomics datasets (*Figure 4a*, *Figure 4—figure supplement 1*). We identified MCM2 as a candidate for further investigation as it was located downstream of ATR and was involved in cell cycle regulation. We then performed western blot and verified that the phosphorylation of MCM2 S108 decreased upon treatment with Arbidol in a dose-dependent manner (*Figure 4b*). Variations in DNA replication timing are often accompanied by changes in the cell cycle (*Prioleau and MacAlpine, 2016*). We used propidium iodide staining to examine the effect of Arbidol on cell cycle progression in KYSE150 and KYSE450 cells after treatment for 48 hr. Results showed that Arbidol strongly reduced the S-phase fraction and induced cell cycle arrest at the G1-phase in a dose-dependent manner (*Figure 4c*). Cyclin-dependent kinases (CDKs) play a primary role in tumor carcinogensis by dysregulating cell cycles. CDK2, CDK4 and CDK6 are expressed during interphase. CDK4 and CDK6 drive cell-cycle progression from G0 or G1-phase into S-phase. CDK2 controls the G1/S checkpoint of the cell cycle (*Malumbres and Barbacid, 2009*). To exclude that the G1-phase cycle block was triggered by the inhibition of the three interphase CDKs, we performed pull-down assays to investigate the potential binding between Arbidol and CDK2, CDK4, or CDK6 (*Figure 4—figure supplement 2a-c*). Our results indicated that Arbidol did not bind with CDK2, CDK4, or CDK6 kinase. This strongly suggested that Arbidol specifically binds with ATR.

RB1, as a tumor suppressor, canonically regulates cell cycle progression and represents a downstream target for CDK4/6 or CDK2 inhibitors that are in clinical use (*Knudsen et al., 2019*). CDKs have been shown to initiate phosphorylation and inactivation of RB1, resulting in derepression of E2F-regulated genes. Phosphorylation of residues within the central pocket and c-terminal domains of RB1 (S249, T252, S356, S373, S608, S612, S780, S788, S795, S807, S811, T821, and S826) have been extensively studied. RB1 S249 is phosphorylated by CDK2, RB1 S807 and RB1 T826 is phosphorylated by CDKs to promote cell cycle progression (*Janostiak et al., 2022*). Therefore, to verify that Arbidol triggers the G1-phase blockade independent of the CDKs/RB signaling pathway, phosphorylated RB1 (S204, S807, and T826) abundance was measured by western blot in cells after Arbidol treatment. As expected, the levels of phosphorylated RB1 (S204, S807, and T826) were not altered after Arbidol treatment (*Figure 4—figure supplement 2d*). These findings demonstrated that Arbidol induced G1-phase block through targeting ATR.

## ATR knockdown reduces the growth of ESCC cells

Next, we determined the correlation between ATR and MCM2 expression using the Bioanalysis website (https://www.aclbi.com/static/index.html#/). The results showed a high correlation between ATR and MCM2 transcript expression (*Figure 5a*). We next queried The Cancer Genome Atlas (TCGA) database (*Tomczak et al., 2015*) to analyze the transcript expression profile of ATR in esophageal cancer and normal tissues based on sample types or tumor history (*Figure 5b*). We also employed Timer 2.0 to analyze the expression status of ATR across tissue samples (*Figure 5c*). These results supported that high expression of ATR correlated with esophageal cancer. To determine the effect of ATR knockdown (KD) on the growth of esophageal cancer cells (KYSE150 and KYSE450), we established ATR stable KD cells and confirmed the expression of ATR protein by western blot. The results showed that the expressions of total ATR in sh *ATR*#1 and sh *ATR*#2 cells were greatly reduced (*Figure 5d*). We then examined the effect of ATR KD on the growth of esophageal cancer cells using a proliferation assay. The results showed that ATR KD significantly reduced the growth of ESCC cells (*Figure 5e and f*). To determine whether the growth reduction of ESCC cells observed upon treatment of Arbidol depended on the expression of ATR, we performed a proliferation assay using ATR KD cells treated with various concentrations (0, 2.5, 5, 10 and 20 µM) of Arbidol. The results indicated the inhibitory

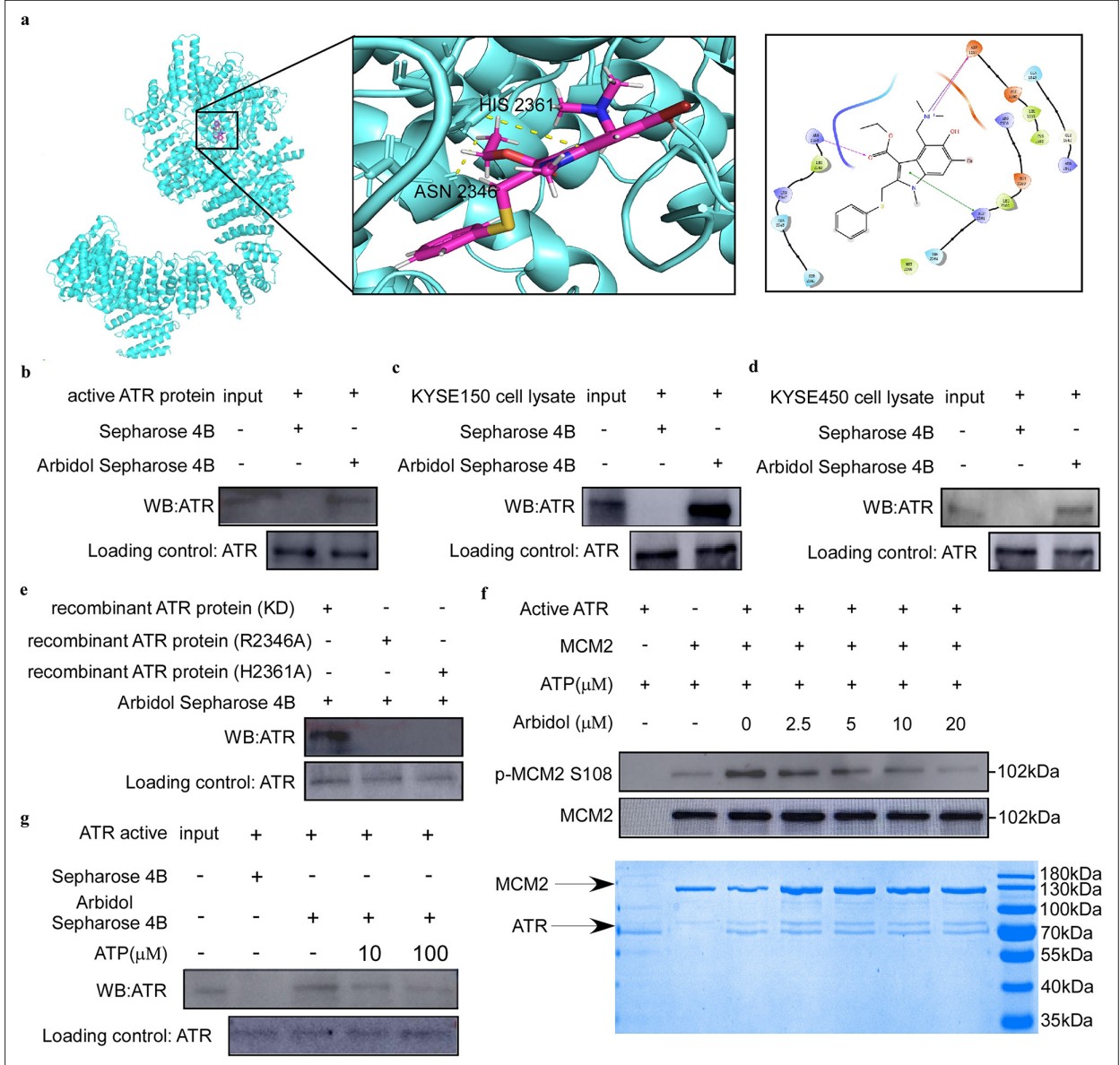

**Figure 3.** Arbidol targets ataxia telangiectasia and Rad3-related (ATR) protein to affect the DNA replication pathway in ESCC. (**a**) Computational docking was used to identify the binding site of Arbidol and ATR. (**b-d**) Arbidol directly binds to ATR. The ATR pure protein (1 μg) (**b**) and esophageal squamous cell carcinoma (ESCC) cell lysate (500 μg) (**c**, **d**) were incubated with sepharose 4B beads conjugated with Arbidol or sepharose 4B beads alone. Western blot analysis was used to analyze the pull-down protein. (**e**) The Arbidol pull-down assay and amino acid site mutation of ATR protein. (**f**) In vitro kinase assay. Arbidol attenuated the phosphorylation of minichromosome maintenance 2 (MCM2) by inhibiting the kinase activity of ATR. Purified MCM2 protein was used as the substrate for the in vitro kinase assay with 200 ng active ATR (SignalChem) at room temperature for 15 min. (**g**) ATP competition assay. ATR pure protein (1 μg) was incubated for 16 hr with sepharose 4B beads conjugated with Arbidol, and with ATP (10 μg or 100 μg) or sepharose 4B beads alone. Western blot analysis was used to analyze the pull-down protein.

The online version of this article includes the following source data and figure supplement(s) for figure 3:

**Source data 1.** The original data of western blot in *Figure 3*.

**Figure supplement 1.** Purified MCM2 and ATR protein.

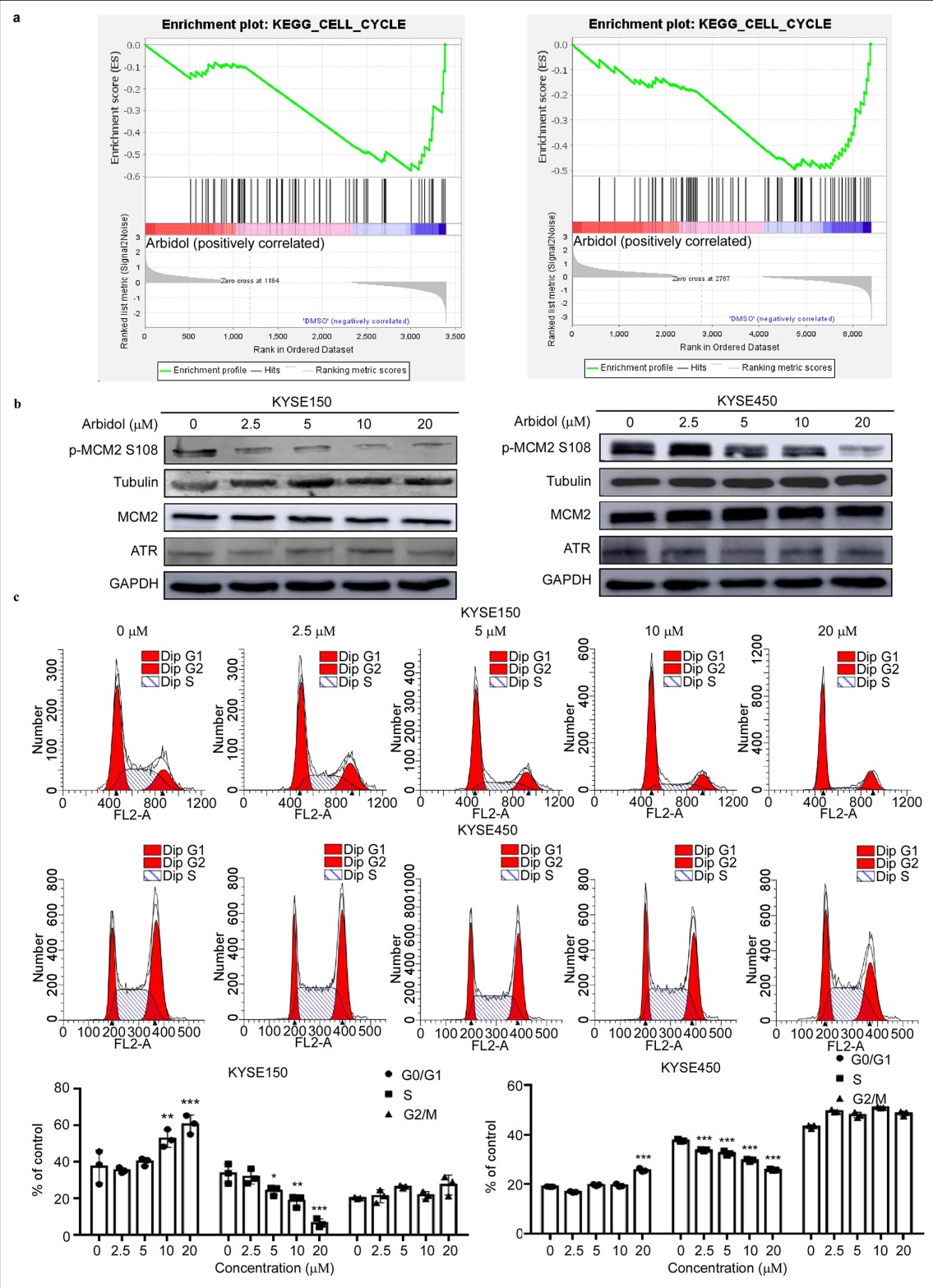

**Figure 4.** Arbidol inhibits the proliferation of ESCC cells and arrests the cell cycle in G1-phase. (**a**) Gene Set Enrichment Analysis (GSEA) indicates the representation of the cell cycle pathway in both the phosphoproteomics and proteomics datasets. (**b**) The expression level of p-minichromosome maintenance 2 (MCM2) S108 decreases with increasing concentrations of Arbidol in KYSE150 and KYSE450 cells (0, 2.5, 5, 10, and 20 μM), as confirmed by western blot. (**c**) After 24 hr of Arbidol treatment, KYSE150 and KYSE450 cells were stained with propidium iodide (PI) and the cell cycle distribution

*Figure 4 continued on next page*

*Figure 4 continued*

was analyzed using flow cytometry. The quantitative cell cycle distribution data of KYSE150 cells were shown here. Data were analyzed by homogeneity of variance and one-way analysis of variance (ANOVA). *p<0.05, ** p<0.01, and ***p<0.001, n=3. ESCC: esophageal squamous cell carcinoma.

The online version of this article includes the following source data and figure supplement(s) for figure 4:

**Source data 1.** The original data of western blot in *Figure 4*.

**Figure supplement 1.** Gene set enrichment analysis (GSEA) data.

**Figure supplement 2.** Arbidol can not bind directly to CDK2,CDK4, CDK6 and can not decrease the levels of p-RB1 S249, S807 and T826.

**Figure supplement 2—source data 1.** The original data of western Blot in *Figure 4—figure supplement 2*.

of effect Arbidol in ATR KD cells (KYSE 150 and KYSE 450) was reduced in a dose dependent manner compared to mock. These results indicated that the inhibitory effect of Arbidol was dependent on ATR (*Figure 5g*).

## Arbidol inhibits tumor growth of ESCC patient-derived xenograft in vivo

To test whether Arbidol exerts an inhibitory effect on ESCC cells in vivo, we utilized an ESCC patient-derived xenograft (PDX) model (*Figure 6a*). The results of the experiment showed that Arbidol treatment effectively reduced tumor volume and weight of EG 20 xenografts (control group, n=7; low Arbidol group, n=7; high Arbidol group, n=7; *Figure 6b*). Importantly, no significant difference in body weights was observed between the control and experimental groups (*Figure 6c-f*). Compared with the control group, histological staining confirmed that Arbidol substantially impaired MCM2 S108 expression in the case of EG20 models, and reduced the expression levels of the Ki67 cell proliferation marker (*Figure 6g*).

## Discussion

The 5-year survival rate of ESCC is less than 20% due to limited preventive methods after primary treatment. Chemoprevention is a promising strategy to reduce the recurrence of ESCC. Several NSAIDs and antimetabolites drugs have been shown to influence the recurrence rate of ESCC. However, few drugs have been successfully repurposed to clinically treat cancer (*Xie et al., 2021*). In the present study, we found that Arbidol had higher IC50 in immortalized esophageal epithelium cell line SHEE than in ESCC cells, which indicated that Arbidol had stronger toxicity in ESCC cells (*Figure 1b and c*). Furthermore, Arbidol significantly reduced ESCC cell proliferation, anchorage-dependent growth, and anchorage-independent growth in vitro (*Figure 1d, e and f*). Additionally, we used a PDX model to evaluate the inhibitory effect of Arbidol in vivo. Our results indicated that Arbidol treatment resulted in a significant reduction of tumor sizes and weights compared to the vehicle-treated group. Importantly, no significant differences in body weights were observed between the treatment and control groups, suggesting that Arbidol could be considered a promising chemoprevention drug against ESCC.

ATR is a key component of the targeted DNA damage response (DDR) (*Jackson and Bartek, 2009*), a new cancer treatment pathway which has broad prospects for tumor selectivity (*Shi et al., 2018*). ATR can phosphorylate proteins involved in DNA replication, DNA recombination and repair, and cell cycle regulation in response to DNA damage (*Snedeker et al., 2017*). Due to its extensive role in DDR, ATR inhibition has great potential in cancer treatment (*Cheng et al., 2017*; *Jin et al., 2021*; *Lu et al., 2020*). Therefore, suppressing ATR signal transduction by inhibitors may exert antitumor effects against ESCC (*Gorecki et al., 2020*).

DNA replication occurs in all organisms that use DNA as their genetic material and their molecular basis of biological inheritance (*Rozpędek et al., 2019*). During DNA replication, a key component called the MCM complex binds to related proteins and provides the helicase activity that unwinds DNA at the origin of replication (*Charrier et al., 2011*; *Kawamura et al., 2019*). The MCM complex undergoes multisite phosphorylation to initiate replication during the transition from G1 to S-phase (*Fei and Xu, 2018*). MCM2, a component of the MCM complex, is required for the initiation and elongation stages of DNA replication; it is also a crucial target of the S-phase-promoting kinases (*Montagnoli et al., 2006*). Phosphorylation of MCM2 on S108 by ATR plays key role in the activation

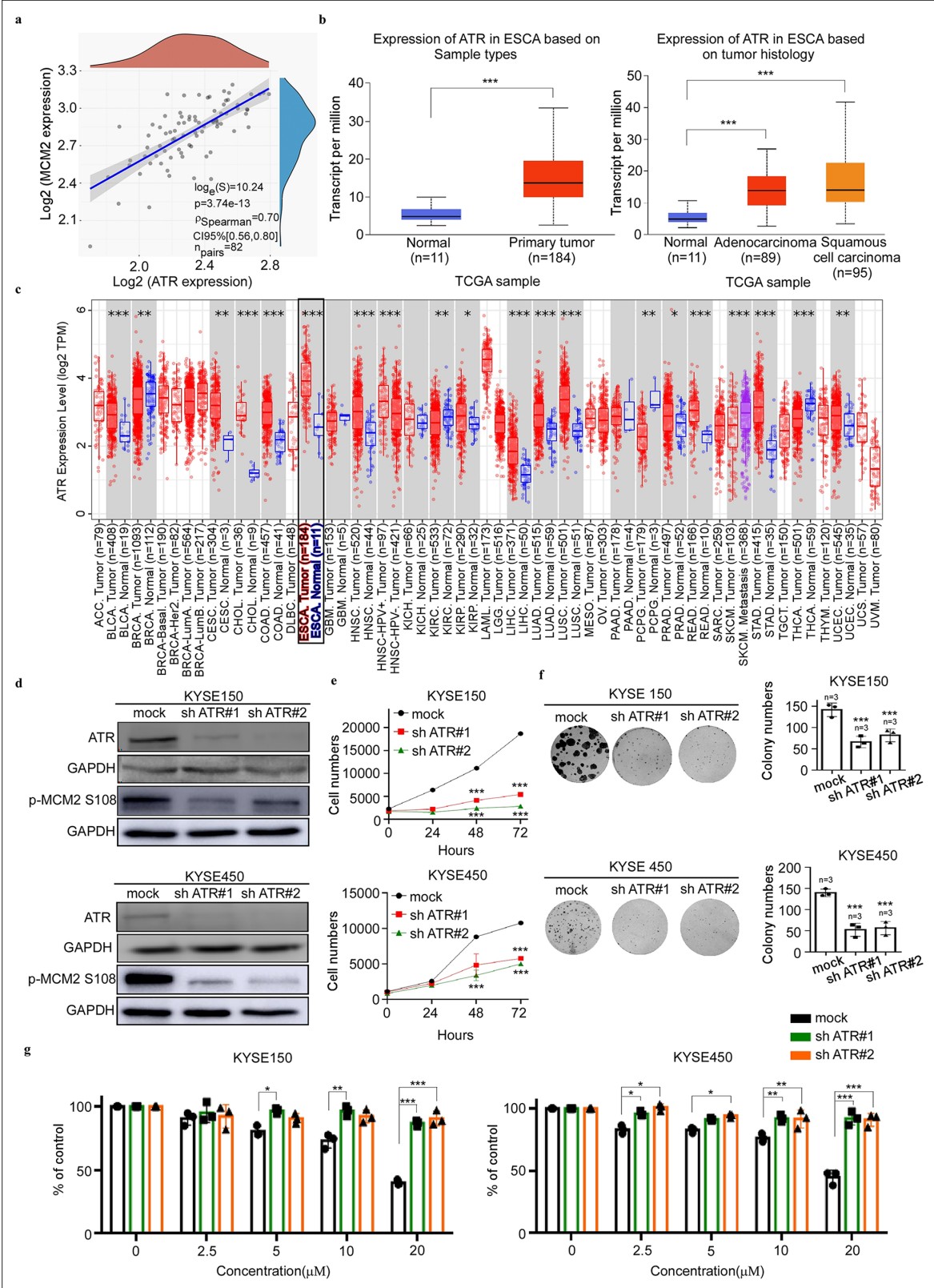

**Figure 5.** Knocking down of ATR reduces the growth of ESCC cells. (**a**) The correlation between ataxia telangiectasia and Rad3-related (ATR) and minichromosome maintenance 2 (MCM2) was detected by gene correlation analysis. (**b**) The Cancer Genome Atlas data results showing the expression of ATR in different cancer types, including esophageal cancer and normal tissue, as well as the statistical chart illustrating esophageal cancer classification. Statistical analyses were performed using R software v4.0.3 (R Foundation for Statistical Computing, Vienna, Austria). p-value of <0.05

*Figure 5 continued on next page*

*Figure 5 continued*

was considered statistically significant. (**c**) The picture is from the results of bioassay on the association between ATR and pan-cancer on Timer 2.0, which includes esophageal cancer. Statistical analyses were performed using R software v4.0.3 (R Foundation for Statistical Computing, Vienna, Austria). The rank sum test detected two sets of data and a p-value of <0.05 is considered statistically significant. (**d**) KYSE150 and KYSE450 cell lines stably expressing knockdown (KD) sh*ATR* or mock were established. The expression of ATR and MCM2 S108 is confirmed by western blot. (**e**) The effect of ATR gene KD on the growth of esophageal cancer cells. The stable KD sh *ATR* cells were seeded into 96-well plates, fixed at 0, 24, 48, 72 hr, and stained with 4',6-diamidino-2-phenylindole, and analyzed by IN cell analyzer. Data were analyzed by homogeneity of variance and one-way analysis of variance (ANOVA). ***p<0.001, n=3. (**f**) The effect of ATR gene KD on the clonogenic potential of esophageal cancer cells. KYSE150 (200/well) and KYSE450 (400/well) cells were seeded on a 6-well plate. The number of cell colonies were counted after 15 days. Data were analyzed by homogeneity of variance and one-way analysis of variance (ANOVA). Asterisks indicate a significant difference, ***p<0.001, n=3, compared with the control group. (**g**) The effect of Arbidol on the growth of esophageal squamous cell carcinoma (ESCC) cells was evaluated in cells stably expressing sh *ATR* or cells stably expressing mock. After inoculating the cells for 12 hr, they were in the presence or absence of different concentrations of Arbidol, and analyzed with IN Cell Analyzer 96 hr later. Data were analyzed by homogeneity of variance and one-way analysis of variance (ANOVA). *p<0.05, ***p<0.001, ***p<0.001, n=3.

The online version of this article includes the following source data for figure 5:

**Source data 1.** The original data of western blot in *Figure 5*.

of prereplication complexes and to stabilize replication forks in response to replication stress (*Thakur et al., 2022*). Phosphorylation of MCM2 S108 by ATR, a mediator of the S-phase checkpoint, is increased upon stalled S-phase (*Cortez et al., 2004*). According to our findings, Arbidol treatment of ESCC cells resulted in decreased S-phase progression and G1-phase blockade. We further verified that Arbidol specifically bound with ATR and not with CDKs. We speculated that the S-phase ESCC cell fraction decreased followed by an increase in the G1-phase fraction after the phosphorylation levels of MCM2 S108 decreased. Therefore, Arbidol decreased the S-phase fraction of ESCC cells and initiated G1-phase blockade through the ATR/MCM2 signaling pathway.

In the present study, we identified that ATR is an upstream kinase of MCM2 using a predictive model based on multiple proteomic assays. ATR was found to be a target of Arbidol through pull-down and in vitro kinase assays (*Figure 3b-f*). We determined that Arbidol bound to ATR at ASN 2346 and HIS 2361 sites (*Figure 3a*), and that these residues were functionally important binding sites for the Arbidol and ATR interaction (*Figure 3e*). Additionally, these residues downregulated the phosphorylation level of MCM2 (*Figure 4b*) and affected ATR kinase activity, as well as the cell cycle (*Figure 4c*).

According to our research, we found that Arbidol is a potential ATR inhibitor that effectively inhibits ESCC growth in vitro and in vivo. At present, the antitumor activity of ATR inhibitors has been observed in many preclinical studies, and the encouraging results of these studies have prompted clinicians to evaluate their effectiveness and safety in clinical trials (*Leszczynska et al., 2016*). Existing ATR inhibitors, such as AZD6738, VX-970/M6620 and BAY-1895344 are currently being tested in a series of clinical trials. Of the listed inhibitors, VX-970 is currently being investigated in a Phase I dose-escalation safety study for treating esophageal cancer and other solid cancers in combination with radiotherapy and chemotherapy (*Shi et al., 2018*). However, the development of effective and specific drugs that target ATR is proceeding slowly (*Gavande et al., 2016*). Compared with the development strategy of de novo synthesis, FDA-approved drugs usually provide pharmacokinetic and toxicity data and are more convenient for clinical translation.

In the present study, we verified the antitumor effects of Arbidol in vivo and in vitro; however, several factors should be further investigated prior to proceeding into the clinical setting. First, a greater number of ESCC cell lines and tissue samples should be used to verify the antitumor effects of Arbidol. PDX models have been shown to mimic several features of human cancer. Within the present study, a subcutaneously implanted tumor model is used. Nonetheless, it may be more ideal if an orthotopic transplantation tumor model is used in future studies. Next, immune responses within the model are neglected as severe combined immune deficiency (SCID) mice are used in ESCC PDX model. Finally, the antitumor efficacy of Arbidol is solely assessed in the in vivo experiments. However, additional preclinical experiments and clinical experiments are necessary for safety evaluation as an anticancer drug.

In summary, our research showed that Arbidol targets ATR, thereby attenuating ESCC proliferation in vitro and in vivo.

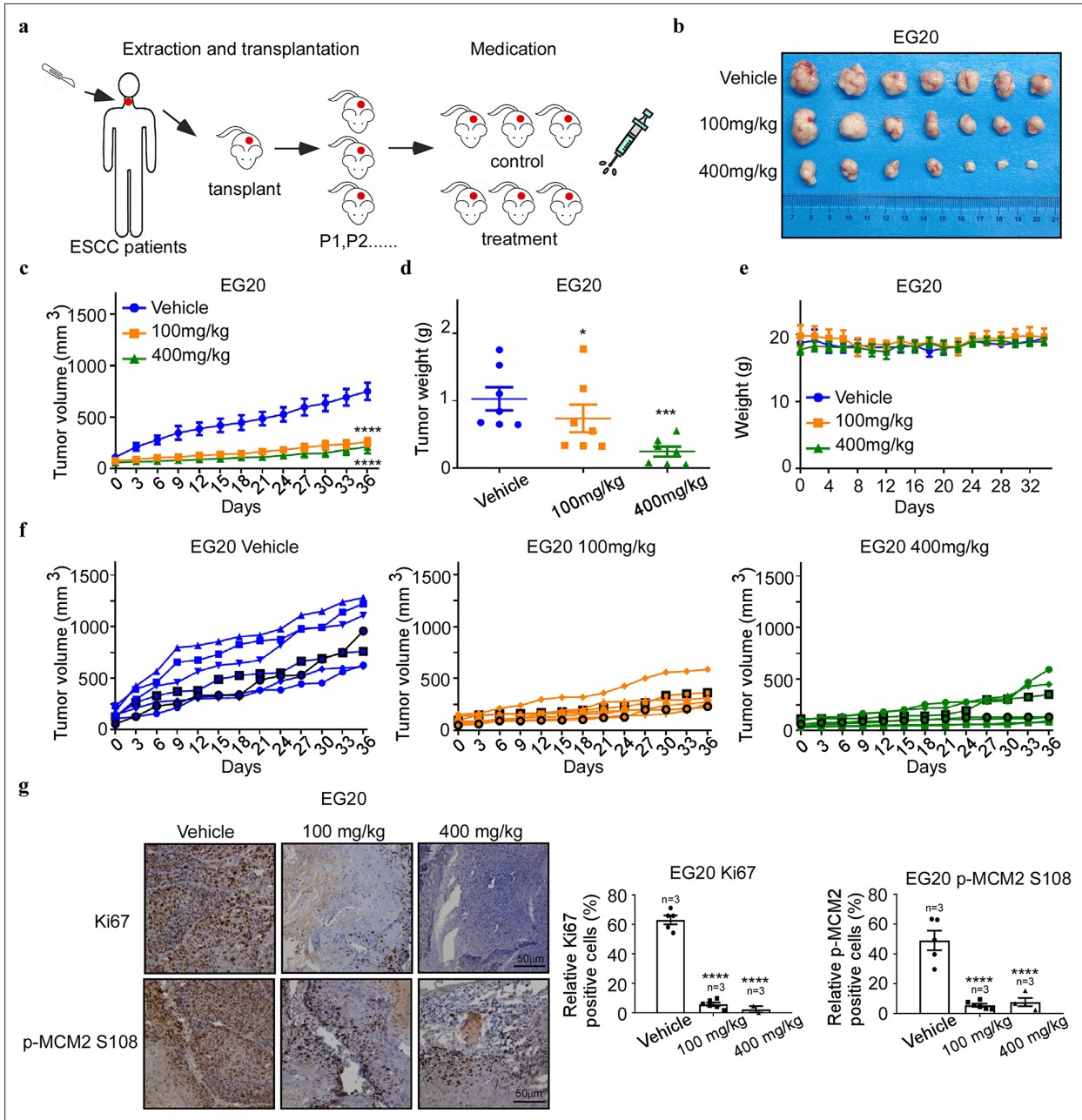

**Figure 6.** Antitumor efficacy of Arbidol in an ESCC patient-derived xenograft model. (**a**) Arbidol treatment protocol for esophageal squamous cell carcinoma (ESCC) patient-derived xenograft (PDX) models. (**b**) Tumor sizes of the EG20 xenografts were shown. Tumors were excised and weighed at the end of the experiment (36 days after treatment). (**c**) The effect of Arbidol inhibition on in vivo tumor growth was determined using ESCC PDX models: EG20. Mice (n=7 per group) were treated with vehicle (saline, oral administration, daily) or Arbidol (low = 100 mg/kg, high = 400 mg/kg, daily) for 36 days. Tumor volumes were measured every 3 days. Data were analyzed by homogeneity of variance and one-way analysis of variance (ANOVA). Data are presented as the mean, ****p<0.0001. (**d**) Tumor growth curve of single mouse grafted with EG20 is shown. n=7 per group. (**e**) Weight of mice treated with Arbidol. Data were analyzed by homogeneity of variance and one-way analysis of variance (ANOVA). *p<0.05, ***p<0.001. (**f**) Tumor volume of each group was shown, respectively, n=7 per group. (**g**) The expression of tumor proliferation markers Ki67 and target engagement was verified by immunohistochemical analysis of minichromosome maintenance 2 (MCM2) S108 expression in Arbidol-treated EG20 PDX mice, n=7 per group. Data were analyzed by homogeneity of variance and one-way ANOVA. *p<0.05, ***p<0.001, ****p<0.0001. Scale Bars: 50 μm.

# Materials and methods

## Reagents

Arbidol was obtained from Beijing Ruitaibio Technology Co, Ltd (AT3360). ATR protein was purchased from SignalChem (No. A27-35G). Anti-rabbit ATR antibody was purchased from Cell Signaling Technology (13934 S). Anti-rabbit MCM2 (phosphoS108) antibody was purchased from Abcam (ab109271). Anti-rabbit MCM2 antibody was purchased from Abcam (ab108935).

## Cell lines and cell culture

Human immortalized esophageal epithelium cell line SHEE cells were gifted from Professor Enmin Li of Shantou University. 293T cells and the human KYSE150 and KYSE450 ESCC cell lines were obtained from Cell Bank of the Chinese Academy of Sciences (Shanghai, China). The KYSE150 cells were maintained in RPMI-1640 medium supplemented with 10% fetal bovine serum (FBS) and 1% penicillin/streptomycin. KYSE450 cells were maintained in RPMI-1640/DMEM supplemented with 10% FBS and 1% penicillin/streptomycin. The 293T cells were maintained in RPMI-DMEM supplemented (10% FBS). Cell lines were maintained at 37°C under 5% $CO_2$ in a humidified incubator. Each cell line was authenticated and periodically monitored for mycoplasma contamination.

## Cell toxicity assay

KYSE150 ($8.0 \times 10^3$ cells/well) and KYSE450 ($1.0 \times 10^4$ cells/well) cells were seeded into 96-well plates with 100 µL of complete growth medium (with 10% FBS) containing various concentrations of Arbidol (final concentration 0, 6.25, 12.5, 25, 50 and 100 µM) and incubated for 24 and 48 hr. The cells were fixed with 100 µL of 4% paraformaldehyde at room temperature for 30 min. Next, 100 µL of 4′,6-diamidino-2-phenylindole (DAPI) solution was added to each well and the cells were incubated in the dark at 37°C for 20 min. The DAPI solution was subsequently discarded and the cells were washed twice with 1×phosphate buffered saline (PBS). Each well was then filled with 100 µL of fresh 1×PBS and high-content cell imaging analysis was used to photograph and analyze the number of cells (*Yuan et al., 2019*). The half-maximal inhibitory concentrations (IC50) values were calculated by SPSS 24.0 software (IBM, USA).

## Cell proliferation assay

KYSE150 ($3.0 \times 10^3$ cells/well) and KYSE450 ($4.0 \times 10^3$ cells/well) cells were seeded into 96-well plates with 100 µL of complete growth medium (10% FBS) containing various concentrations of Arbidol (final concentration 0, 2.5, 5, 10, and 20 µM) and incubated at 37°C and 5% $CO_2$ for 24, 48, 72, and 96 hr. The cells were fixed with 4% paraformaldehyde at room temperature for 30 min. Next, 100 µL of DAPI solution was added to each well and the cells were incubated in the dark at 37°C for 20 min. The DAPI working solution was subsequently discarded and the cells were washed twice with 1×PBS. Each well was then filled with 100 µL of fresh 1×PBS and high-content cell imaging analysis was used to photograph and analyze the number of cells (*Yuan et al., 2019*).

## Soft agar assay

Basal medium eagle (Sigma-Aldrich, UK) supplemented with 10% FBS, 0.5% agarose, 9% sterile water, 1% glutamine, 0.1% gentamicin, and various concentrations of Arbidol (final concentration 0, 2.5, 5, 10, and 20 µM) were added to each well of a 6-well plate, at 3 mL per well. After 2 hr, cells ($8 \times 103$ per well) were suspended in 10% FBS, 0.33% agar, 45% sterile water, 1% glutamine, 0.1% gentamicin, and various concentrations of Arbidol (0, 2.5, 5, 10, and 20 µM). The cell mixtures were then applied over the solidified bottom layer. The cells were incubated at 37°C and 5% $CO_2$ for 8 days. Afterward, the number of colonies was counted using an In CellAnalyzer 6000.

## Colony formation assay

Cells were plated into 6-well plates at 200 cells per well and allowed to adhere for 12–16 hr. Upon adherence, the cells were treated with various concentrations of Arbidol (0, 2.5, 5, 10, and 20 µM) for 10–14 days. The cell colonies were fixed with 4% triformol, stained with 0.1% crystal violet for 5 min, and then the number of colonies were counted.

## Western blot analysis

Cells were treated with various concentrations of Arbidol for 24 hr. The culture medium was then discarded and the cells were washed with 1×PBS prior to collection in radio immunoprecipitation buffer supplemented with phenylmethylsulfonyl fluoride and phosphatase inhibitor cocktail. Cell lysates were incubated for 1 hr on ice, sonicated, and centrifuged for 30 min at 14,000 rpm and 4°C. Protein concentrations were determined using the bicinchoninate method. The protein samples were resolved via sodium dodecyl sulfate polyacrylamide gel electrophoresis and transferred onto polyvinylidene difluoride membranes in transfer buffer. The membranes were incubated with the indicated antibodies prior to detection using enhanced chemiluminescent reagent.

## Pull-down assay

Activated beads were suspended with 1 mM HCl, and a 40% dimethyl sulfoxide/H2O (v/v)-coupling buffer (0.5 M NaCl, pH 8.3) and then mixed with Arbidol (4 mg) prior to rotation overnight at 4°C. The beads were transferred to 0.1 M Tris-HCl buffer (pH 8.0) and again rotated overnight at 4°C. Finally, the beads were washed three times with 0.1 M acetate buffer (pH 4.0) containing 0.5 M NaCl followed by one additional wash with 0.1 M Tris-HCl (pH 8.0) containing 0.5 M NaCl (*Steele et al., 2003*).

## Molecular docking assay

The 2D structure of Arbidol (PubChem CID: 131411) was downloaded from the PubChem website (https://pubchem.ncbi.nlm.nih.gov/). The crystal structure of ATR (5YZ0) was downloaded from the Protein Database (PDB) website (https://www.rcsb.org/). The water molecules and small molecule ligands in the structure were deleted prior to the addition of hydrogen atoms. Computational docking between Arbidol and ATR was performed using the AutoDock 4.2.6 software. The optimal complex was visualized using PyMOL (PyMOL molecular graphics system, version 2.3.4).

## In vitro kinase assay

MCM2 was the substrate ATR for the in vitro kinase assay, 200 ng active ATR (Signalchem) and purified MCM2 protein were incubated in 1×kinase buffer (25 mM Tris-HCl pH 7.5, 5 mM β-glycerophosphate, 2 mM DTT, 0.1 mM Na3VO4, 10 mM MgCl2, and 5 mM MnCl2) containing 200 μM ATP at 30°C for 30 min. The reaction was terminated by adding 6×protein loading buffer. The protein was detected by western blot.

## Plasmids, transfection, and infection

ATR-specific short hairpin RNAs (shRNAs) were purchased from Sangon Biotech and then cloned into the pLKO.1 vector. The hairpin sequences used were as follows: sh *ATR*#1: 5'-GCCGCTAATCTTCTAACATTA-3' sh *ATR*#2: 5'-TAATGGTTCTTACTCGTATTA-3'. To prepare ATR lentivirus particles, we used jetPRIME to transfect each viral plasmid and packaging plasmids (pMD2.G, psPAX2) into HEK293T cells according to the protocol recommended by the manufacturer. After 48 hr of incubation, the virus-enriched medium was collected and passed through a 0.45 μm sodium acetate filter. The viral medium was then supplemented with 8 μg/mL polyethylene and used to infect KYSE150 or KYSE450 cells. After 24 hr of incubation, cells were selected with puromycin (1 μg/mL) for 48 hr.

## PDX model

In accordance with the guidelines of the Ethics Committee of Zhengzhou University (Zhengzhou, Henan, China), 6–8 week old severe combined immunodeficiency (SCID) female mice (Vital River Labs, Beijing, China) were used for patient-derived xenograft (PDX) model experiments. The EG 20 ESCC case was used for the PDX animal experiments. The tumors were divided into 0.1–0.2 g fragments and subcutaneously inoculated into the backs of SCID mice. The treatment regimens were initiated approximately two weeks after tumor inoculation. The mice were randomly divided into the following groups: control group, 100 mg/kg Arbidol treatment group, and 400 mg/kg Arbidol treatment group. Arbidol or dimethyl sulfoxide was administered to each mouse daily by oral gavage. The tumor volume and weight of each mouse were measured every 3 days. The tumor volumes were calculated according to the size of a single tumor and the following formula: tumor volume (mm3)=1/2×length × width2 (*Xie et al., 2021*).

## Cell cycle assay

KYSE150 and KYSE450 cells were cultured in 6-well plates at a density of 8×105 and 1.2×106 cells/well, respectively. After serum starvation for 24 hr, the cells were treated with Arbidol (0, 2.5, 5, 10, and 20 µM). The cells were then harvested and fixed in 70% cold ethanol at 4°C overnight. After fixation, the cells were washed three times with PBS, stained with propidium iodide solution for approximately 30 min at room temperature, and analyzed using a FACScan flow cytometer. Data were analyzed with FlowJo software (Tree Star, Inc).

## Liquid chromatography tandem mass spectrometry (lc-ms/ms) analysis

The peptide was dissolved in solvent A (0.1% formic acid and 2% acetonitrile in water) and separated using an ultra-high-performance liquid chromatography (UPLC) system (EASY-nLC 1000; Thermo Scientific). The gradient included an increase from 6 to 22% solvent B (0.1% formic acid in 90% acetonitrile) for 40 min, 22 to 35% solvent B for 12 min, 35 to 80% solvent B for 4 min, and rising to 80% solvent B for the last 4 min. A constant flow rate of 700 nL/min was used. The separated peptides were then ionized using a nanospray ionization ion source and analyzed using an Orbitrap Fusion (Thermo Scientific) MS. High-resolution Orbitrap was used to detect and analyze the precursor ions and secondary fragments of peptides. The m/z scan range of the full scan was 350–1550, and the complete peptide was detected in the Orbitrap at a resolution of 60,000. The scanning range of the secondary mass spectrometer was fixed at the starting point of 100 m/z, and the resolution was set to 15,000. After a full scan, the top 20 peptides with the highest signal intensity were selected to enter the high-energy collision dissociation library and fragmented using 35% of the fragmentation energy. A secondary MS was also performed.

## Gene set enrichment analysis ( GSEA) for data processing

The GSEA v4.0.3 software package was used to integrate the proteome data set with Gene Set Enrichment Analysis (GSEA) and determine the difference in protein expression between the treatment and control groups. The mitochondrial protein complex genome (GO: 0098798) and the respiratory electron transport genome (R-HSA-611105) were selected for enrichment. Through previous functional annotations or experiments, GSEA determined the relevant gene set and sequenced the genes based on the degree of differential expression of the two samples. Gene set enrichment analysis detected changes in gene set expression and optimized, the results using p-value and false discovery rate (FDR) methods. Biological pathways with p<0.01 and FDR<0.25 were considered statistically significant.

## Immunohistochemical (IHC) analysis

The IHC assay was carried out as described previously (*Xie et al., 2021*). Briefly, paraffin-embedded tissues (5 µm) were prepared for IHC analysis. The specimens were baked in a constant-temperature oven at 65°C for 2 hr, de-paraffinized, and rehydrated prior to antigen retrieval. All samples were treated with 3% $H_2O_2$ for 10 min in the dark to inactivate endogenous peroxidases. The slides were then incubated with specific antibodies overnight at 4°C. The tissue sections were washed three times with 1× tris-buffered saline with 0.1% Tween 20 detergent and hybridized with the secondary antibody for 15 min at 37°C. After 3, 3'-diaminobenzidine (DAB) staining, all slides were stained with hematoxylin, dehydrated and mounted under glass coverslips.

## Statistical analysis

The statistical significance of results was assessed using SPSS 24.0 software (IBM, USA) or R software v4.0.3 (R Foundation for Statistical Computing, Vienna, Austria). All quantitative results are expressed as mean ± SD. Statistically significant differences were calculated using the Student's t-test or one-way ANOVA. p<0.05 was considered statistically significant. *p<0.05, **p<0.01, and ***p<0.001.

## Acknowledgements

Thanks to Xiaofan Zhang, Xiangyu Wu, Zhuo Bao, Bo Li and Yubing Zhou from Pathophysiology Department, Zhengzhou university. This research was supported by the National Natural Science Foundation of China (grant number 81872335), the National Natural Science Youth Foundation (grant

number 81902486), and the Natural Science Foundation of Henan (grant number 161100510300), The Central Plains Science and Technology Innovation Leading Talents (grant number 224200510015).

## Additional information

### Funding

| Funder | Grant reference number | Author |
|---|---|---|
| National Natural Science Foundation of China | 81872335 | Kangdong Liu |
| National Natural Science Youth Foundation of China | 81902486 | Yanan Jiang |
| Natural Science Foundation of Henan Province | 161100510300 | Kangdong Liu |
| The Central Plains Science and Technology Innovation Leading Talents | 224200510015 | Kangdong Liu |

The funders had no role in study design, data collection and interpretation, or the decision to submit the work for publication.

### Author contributions

Ning Yang, Data curation, Software, Formal analysis, Writing - original draft, Did the pull down assay, knock down assay, western Blot Analysis, In vitro Kinase Assay, process the data and so on; Xuebo Lu, Data curation, Software, Writing – review and editing, Did the animal assay and molecular docking assay; Yanan Jiang, Conceptualization, Resources, Data curation, Formal analysis, Funding acquisition, Validation, Investigation, Visualization, Methodology, Writing – review and editing; Lili Zhao, Software, Immunohistochemical (IHC) analysis and cell cycle assay; Donghao Wang, Did the animal assay and processed the data for Gene Set Enrichment Analysis (GSEA); Yaxing Wei, Did most of in vitro assay; Yin Yu, Software, Purified protein; Myoung Ok Kim, Methodology, Writing – review and editing; Kyle Vaughn Laster, Writing – review and editing; Xin Li, Resources, Software, Writing – review and editing; Baoyin Yuan, Software, Writing – review and editing; Zigang Dong, Conceptualization, Resources, Validation, Writing – review and editing; Kangdong Liu, Conceptualization, Resources, Data curation, Software, Formal analysis, Supervision, Funding acquisition, Validation, Investigation, Visualization, Methodology, Project administration, Writing – review and editing

### Author ORCIDs

Xuebo Lu http://orcid.org/0000-0001-9481-2568
Kangdong Liu http://orcid.org/0000-0002-4425-5625

### Ethics

Compliance with Ethics Requirements In this study, we established an ESCC PDX model. In this model, the tumor sample from an ESCC patient was EG20 (ESCC, male, T2N0M0II, moderately differentiated, obtained from Linzhou Cancer Hospital, Henan Province, China). The patient was fully informed of the study and provided consent. This study was approved by the Ethics Committee of Zhengzhou University (ZZUHCI-2019012).

### Decision letter and Author response

Decision letter https://doi.org/10.7554/eLife.73953.sa1
Author response https://doi.org/10.7554/eLife.73953.sa2

## Additional files

### Supplementary files
• Transparent reporting form

## Data availability

The mass spectrometry proteomics data have been deposited to the ProteomeXchange Consortium (http://proteomecentral.proteomexchange.org) via the iProX partner repository with the dataset identifier PXD034944.

The following dataset was generated:

| Author(s) | Year | Dataset title | Dataset URL | Database and Identifier |
|---|---|---|---|---|
| Jiang Y, Liu K | 2022 | Proteomics analysis report of esophageal squamous cell carcinoma treated by Arbidol | http://proteomecentral.proteomexchange.org/cgi/GetDataset?ID=PXD034944 | ProteomeXchange, PXD034944 |

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
