## [Editor Report]

This manuscript will be of interest to a broad audience of cancer biologists, especially those interested in esophageal cancer or treatment strategies involving ATR inhibition. It provides novel information about how FDA-approved antiretroviral compound Arbidol is a potential ATR inhibitor, which is of interest in the treatment of multiple tumor types. The key claims of the manuscript are supported by in silico, in vitro, and in vivo data.

---

## [Decision Letter]

**Decision letter after peer review:**

Thank you for submitting your article "Arbidol inhibits esophageal squamous cell carcinoma growth in vitro and in vivo through suppressing ataxia telangiectasia and Rad3-related protein kinase" for consideration by *eLife*. Your article has been reviewed by 2 peer reviewers, and the evaluation has been overseen by a Reviewing Editor and Wafik El-Deiry as the Senior Editor. The following individual involved in review of your submission has agreed to reveal their identity: Haidan Liu (Reviewer #2).

The reviewers have discussed their reviews with one another, and the Reviewing Editor has drafted this to help you prepare a revised submission. Please address the comments of the two reviewers below. Please follow and address all comments below – Recommendations to Authors.

*Reviewer #1 (Recommendation for the Authors):*

– Statistical methods used for each experiment should be detailed in each figure legend and in the 2.17 Statistical Analysis methods section. Additionally, please add sample size (n) values to each of the figure legends, including for in vitro data.

– The authors need to mention the limitations of the study, especially concerning the in vivo experiment.

– Methods of IC-50 determination and the software used should be included in the methods section of the manuscript. The authors should also acknowledge the implications of the high IC-50 of the compound in the ESCC cell lines examined.

– Please clarify treatment dosing- Line 198 states that mice were divided into a "100 mg/kg Arbidol hydrochloride treatment group, and 200-400 mg/kg Arbidol hydrochloride treatment group" but figure 6 only shows 100 and 400 mg/kg groups. Also please state when the daily Arbidol hydrochloride treatment started for the in vivo experiment- was it on the day of tumor inoculation?

– The authors state in Line 329 that their results in figure 5C support the idea that high expression of ATR correlates with decreased overall survival in esophageal cancer, but Figure 5 lacks survival data.

– What was the treatment duration for the assay as described in Figure 3G? 2F? 3F?

– In figure 5E, the legend says cells were fixed at "0 h, 24 h, 48 h, 72 h, and 96 h" but there is no data for the 96-hour timepoint.

– Please show individual data points overlayed on top of the bar graphs for the following panels: 1E, 1F, 4C, 5F, 5G, 6G.

– For Figure 2G please further elaborate on line 286 "The peptide figure showed that the test conformed the standard". Additionally, the font size in 2G is too small to be legible.

*Reviewer #2 (Recommendation for the Authors):*

However, several issues need to be addressed:

1. Since a G1 phase cycle block can be observed and this effect could be also triggered by the inhibition of CDKs, this possibility should be excluded.

2. The authors described that Arbidol targets ATR protein to affect the DNA replication pathway and confirmed by cell cycle analysis. The authors need to give a more thorough discussion on the functional relationship between DNA replication and cell cycle.

3. In Figure 3a, the binding of Arbidol to ATR at ASN 2361 residue is missing. The authors should include this residue.

4. Figure 5g should include a clear description of the experiment performed.

5. What is the p-value for "****" in Figures 6c and 6g?

6. In Figures 1e and 6g, length unit should be "50 μm" but not "50 μM". Typo on line 252: "*P < 0.05, **P < 0.01, ***P < 0.001", "P" should be lower case. Typo on line 484: "P <0.01" should be "**p <0.01". The whole manuscript needs to be carefully checked.

---

## [Author Response]

Reviewer #1 (Recommendation for the Authors):– Statistical methods used for each experiment should be detailed in each figure legend and in the 2.17 Statistical Analysis methods section. Additionally, please add sample size (n) values to each of the figure legends, including for in vitro data.

Thanks for your suggestions. We have added the related information in the 2.17 Statistical Analysis methods section and each figure legend. Sample size (n) values to each of the figure legends were also added to each figure legend.

The information added in the 2.17 Statistical Analysis methods section is as follows: Statistically significant differences were checked using the SPSS 24.0 software (IBM, USA) or R software v4.0.3 (R Foundation for Statistical Computing, Vienna, Austria); ***p < 0.001.

The information to each figure legend is as follows:

Figure 1c: Data were analyzed by homogeneity of variance and one-way analysis of variance (ANOVA). Asterisks indicate a significant decrease. *p < 0.05, **p < 0.01, ***p < 0.001, n=3.

Figure 1d: Data were analyzed by homogeneity of variance and one-way analysis of variance (ANOVA). Asterisks indicate a significant difference. *p <0.05, ** p <0.01, and ***p <0.001, n=3, compared with the control group.

Figure 1e: Data were analyzed by homogeneity of variance and one-way analysis of variance (ANOVA). Asterisks indicate a significant difference. *p <0.05, ** p <0.01, and ***p <0.001, n=3, compared with the control group.

Figure 4c: Data were analyzed by homogeneity of variance and one-way analysis of variance (ANOVA). *p <0.05, ** p <0.01, and ***p <0.001, n=3.

Figure 5b, c: Statistical analyses were performed using R software v4.0.3 (R Foundation for Statistical Computing, Vienna, Austria). A p-value <0.05 was considered statistically significant.

Figure 5e: Data were analyzed by homogeneity of variance and one-way analysis of variance (ANOVA). ***p <0.001, n=3.

Figure 5f: Data were analyzed by homogeneity of variance and one-way analysis of variance (ANOVA). ***p <0.001, n=3.

Figure 5g: Data were analyzed by homogeneity of variance and one-way analysis of variance (ANOVA). *p < 0.05, ***p < 0.001, ***p < 0.001, n=3.

Figure 6c: Data were analyzed by homogeneity of variance and one-way analysis of variance (ANOVA). Data are presented as the mean. ****p < 0.0001.

Figure 6d: Data were analyzed by homogeneity of variance and one-way analysis of variance (ANOVA). *p < 0.05, ***p < 0.001.

Figure 6g: Data were analyzed by homogeneity of variance and one-way analysis of variance (ANOVA). *p < 0.05, ***p < 0.001, ****p < 0.0001.

– The authors need to mention the limitations of the study, especially concerning the in vivo experiment.

Thank you for the suggestion. We have discussed the limitations of the study in the Discussion section. The limitations of the in vivo experiment are also discussed:

“In the present study, we verified the anti-tumor effects of Arbidol in vivo and in vitro. Certain limitations could be overcome for utilizing Arbidol as an anti-cancer drug. First, a greater number of ESCC cell lines and tissue samples should be used to verify the anti-tumor effects of Arbidol. PDX models have been shown to mimic several features of human cancer. Within the present study, a subcutaneously implanted tumor model is used. Nonetheless, it may be more ideal if an orthotopic transplantation tumor model is used in future studies. Next, immune responses within the model are neglected as severe combined immune deficiency (SCID) mice are used in ESCC PDX model. Finally, the anti-tumor efficacy of Arbidol is solely assessed in the in vivo experiments. However, additional pre-clinical experiments and clinical experiments are necessary for safety evaluation as an anti-cancer drug.”

– Methods of IC-50 determination and the software used should be included in the methods section of the manuscript. The authors should also acknowledge the implications of the high IC-50 of the compound in the ESCC cell lines examined.

Thanks for your comments. The IC50 determination methods and the software used are described in Cell Toxicity Assay of the Materials and methods section: IC50 values were calculated by SPSS 24.0 software (IBM, USA).

We also mentioned the higher IC50 value of the compound in the immortalized esophageal epithelium cell line SHEE compared to ESCC cells in the Discussion section: In the present study, we found that Arbidol had a higher IC50 value in the immortalized esophageal epithelium cell line SHEE than in ESCC cells, which indicated that Arbidol had stronger toxicity in ESCC cells (Figure 1b, 1c).

– Please clarify treatment dosing- Line 198 states that mice were divided into a "100 mg/kg Arbidol hydrochloride treatment group, and 200-400 mg/kg Arbidol hydrochloride treatment group" but figure 6 only shows 100 and 400 mg/kg groups. Also please state when the daily Arbidol hydrochloride treatment started for the in vivo experiment- was it on the day of tumor inoculation?

Thank you for pointing out this mistake. We only used 100 and 400 mg/kg Arbidol treatment groups in the experiment. We corrected this information in the Methods section: The mice were randomly divided into the following groups: control group, 100 mg/kg Arbidol treatment group, and 400 mg/kg Arbidol treatment group.

We stated the starting time for the in vivo experiment with Arbidol treatment in the Materials and methods section: The tumors were divided into 0.1-0.2 g fragments and subcutaneously inoculated into the backs of SCID mice. The treatment regimens were initiated approximately two weeks after tumor inoculation.

– The authors state in Line 329 that their results in figure 5C support the idea that high expression of ATR correlates with decreased overall survival in esophageal cancer, but Figure 5 lacks survival data.

Thank you for pointing out this. This sentence is inaccurate. We deleted this sentence in the revised manuscript.

– What was the treatment duration for the assay as described in Figure 3G? 2F? 3F?

Thank you for the questions. We added this information in the figure legends.

The treatment duration for this assay in Figure 3G was 16 hours.

In figure 2G, the protein was collected after the drug treatment for 24 hours. Note that figures 2F and 2G were switched upon manuscript revision.

In figure 3F, purified MCM2 protein was used as the substrate for the in vitro kinase assay with 200 ng active ATR (Signalchem) at room temperature for 15 min.

– In figure 5E, the legend says cells were fixed at "0 h, 24 h, 48 h, 72 h, and 96 h" but there is no data for the 96-hour timepoint.

Thank you for pointing out this mistake, we corrected this information in the figure legend of figure 5E: The sh ATR stable knockdown cells were seeded into 96-well plates and fixed at 0 h, 24 h, 48 h and 72 h.

– Please show individual data points overlayed on top of the bar graphs for the following panels: 1E, 1F, 4C, 5F, 5G, 6G.

Thank you for your suggestion, we overlaid the data points on top of the bar graphs in figures 1E, 1F, 4C, 5F, 5G, and 6G.

– For Figure 2G please further elaborate on line 286 "The peptide figure showed that the test conformed the standard". Additionally, the font size in 2G is too small to be legible.

Thank you so much for your comments. To make the results logical, we switched the figure sequence of Figure 2f and 2g. The sentence was changed to: The peptide profile of MCM2 S108 obtained using mass spectrometry was displayed in Figure 2f.

A clear version of the peptide diagram was produced and added in the revised manuscript.

Reviewer #2 (Recommendation for the Authors):However, several issues need to be addressed:1. Since a G1 phase cycle block can be observed and this effect could be also triggered by the inhibition of CDKs, this possibility should be excluded.

Thanks for your suggestions. Our results showed that Arbidol strongly reduced the S phase fraction and induced cell cycle arrest at the G1 phase in a dose-dependent manner (Figure 4c). Cyclin-dependent kinases (CDKs) play a primary role in tumor carcinogenesis by dysregulating the cell cycle. CDK2, CDK4, and CDK6 are expressed during interphase. CDK4 and CDK6 drive cell-cycle progression from G0 or G1 phase into S phase. CDK2 controls the G1/S checkpoint of the cell cycle (Malumbres and Barbacid, 2009). To exclude that the G1 phase cycle block was triggered by the inhibition of the three interphase CDKs, we performed pull-down assays to investigate potential binding between Arbidol and CDK2, CDK4, or CDK6. Our results indicated that Arbidol did not bind with CDK2, CDK4, or CDK6 kinase. This strongly suggested that Arbidol specifically binds with ATR.

RB1, as a tumor suppressor, canonically regulates cell cycle progression and represents a down-stream target for CDK4/6 or CDK2 inhibitors that are in clinical use (Knudsen et al., 2019). CDKs have been shown to initiate phosphorylation and inactivation of RB1, resulting in derepression of E2-regulated genes. Phosphorylation of residues within the central pocket and c-terminal domains of RB1 (S249, T252, S356, S373, S608, S612, S780, S788, S795, S807, S811, T821, S826) have been extensively studied. RB1 S249 is phosphorylated by CDK2, RB1 S807 and RB1 T826 is phosphorylated by CDKs to promote cell cycle progression (Janostiak et al., 2022). Therefore, to verify that Arbidol triggers the G1 phase blockade independent of the CDKs/RB signaling pathway, phosphorylated RB1 (S204, S807, T826) abundance was measured by western blotting in cells after Arbidol treatment. As expected, the levels of phosphorylated RB1 (S204, S807, T826) were not altered after Arbidol treatment (Fig S1d). These findings demonstrated that Arbidol induced G1 phase block through targeting ATR. We have added these results as Figure S1 in the Results section of the revised manuscript.

2. The authors described that Arbidol targets ATR protein to affect the DNA replication pathway and confirmed by cell cycle analysis. The authors need to give a more thorough discussion on the functional relationship between DNA replication and cell cycle.

Thanks for your suggestions. We further discussed the relationship between DNA replication and cell cycle in the third paragraph of the Discussion section. We also displayed this part here:

“DNA replication occurs in all organisms that use DNA as their genetic material and their molecular basis of biological inheritance (Rozpędek et al., 2019). During DNA replication, a key component called the MCM complex binds to related proteins and provides the helicase activity that unwinds DNA at the origin of replication (Charrier et al., 2011; Kawamura et al., 2019) The MCM complex undergoes multi-site phosphorylation to initiate replication during the transition from G1 to S phase (Fei and Xu, 2018). MCM2, a component of the MCM complex, is required for the initiation and elongation stages of DNA replication; it is also a crucial target of the S-phase-promoting kinases (Montagnoli et al., 2006). Phosphorylation of MCM2 on S108 by ATR plays key roles in the activation of pre-replication complexes and to stabilize replication forks in response to replication stress (Thakur et al., 2022). Phosphorylation of MCM2 S108 by ATR, a mediator of the S phase checkpoint, is increased upon stalled S-phase (Cortez et al., 2004). According to our findings, Arbidol treatment of ESCC cells resulted in decreased S-phase progression and G1-phase blockade. We further verified that Arbidol specifically bound with ATR and not with CDKs. We speculated that the S-phase ESCC cell fraction decreased followed by an increase in the G1-phase fraction after the phosphorylation levels of MCM2 S108 decreased. Therefore, Arbidol decreased the S-phase fraction of ESCC cells and initiated G1-phase blockade through the ATR/MCM2 signaling pathway.”

3. In Figure 3a, the binding of Arbidol to ATR at ASN 2361 residue is missing. The authors should include this residue.

Thank you for your useful suggestion. Due to the different viewpoint in the previous version of the manuscript, Arbidol binding to ATR at HIS 2361 was not sufficiently 2visualized. We have changed the viewpoint and produced a new figure that displays both ASN 2346 and HIS 2361.

4. Figure 5g should include a clear description of the experiment performed.

Thank you for your suggestions. We have given a clear description of how we performed the experiments in the revised manuscript:

“The results indicated the inhibitory effects of Arbidol were reduced in both KYSE 150 sh ATR and KYSE 450 sh ATR cells compared to mock. These results showed the inhibitory effects of Arbidol are dependent on ATR kinase (Figure 5g).”

5. What is the p-value for "****" in Figures 6c and 6g?

Thank you for the question. "****" in Figures 6c and 6g denotes p<0.0001. We also added this information in the Statistical Analysis of Materials and methods section.

6. In Figures 1e and 6g, length unit should be "50 μm" but not "50 μM". Typo on line 252: "*P < 0.05, **P < 0.01, ***P < 0.001", "P" should be lower case. Typo on line 484: "P <0.01" should be "**p <0.01". The whole manuscript needs to be carefully checked.

Thanks for your comments. We have carefully checked all the texts and changed the unclear expressions in the revised manuscript.

References:

Charrier JD, Durrant SJ, Golec JMC, Kay DP, Knegtel RMA, MacCormick S, Mortimore M, O’Donnell ME, Pinder JL, Reaper PM, Rutherford AP, Wang PSH, Young SC, Pollard JR. 2011. Discovery of potent and selective inhibitors of ataxia telangiectasia mutated and Rad3 related (ATR) protein kinase as potential anticancer agents. *Journal of medicinal chemistry* 54:2320–2330. doi:10.1021/JM101488Z

Cortez D, Glick G, Elledge SJ. 2004. Minichromosome maintenance proteins are direct targets of the ATM and ATR checkpoint kinases. Proceedings of the National Academy of Sciences of the United States of America 101:10078–10083. doi:10.1073/PNAS.0403410101

Fei L, Xu H. 2018. Role of MCM2-7 protein phosphorylation in human cancer cells. *Cell & bioscience* 8. doi:10.1186/S13578-018-0242-2

Janostiak R, Torres-Sanchez A, Posas F, de Nadal E. 2022. Understanding Retinoblastoma Post-Translational Regulation for the Design of Targeted Cancer Therapies. *Cancers* 14. doi:10.3390/CANCERS14051265

Kawamura K, Qi F, Meng Q, Hayashi I, Kobayashi J. 2019. Nucleolar protein nucleolin functions in replication stress-induced DNA damage responses. *Journal of radiation research* 60:281–288. doi:10.1093/JRR/RRY114

Knudsen ES, Pruitt SC, Hershberger PA, Witkiewicz AK, Goodrich DW. 2019. Cell Cycle and Beyond: Exploiting New RB1 Controlled Mechanisms for Cancer Therapy. *Trends in cancer* 5:308–324. doi:10.1016/J.TRECAN.2019.03.005

Malumbres M, Barbacid M. 2009. Cell cycle, CDKs and cancer: a changing paradigm. *Nature reviews Cancer* 9:153–166. doi:10.1038/NRC2602

Montagnoli A, Valsasina B, Brotherton D, Troiani S, Rainoldi S, Tenca P, Molinari A, Santocanale C. 2006. Identification of Mcm2 phosphorylation sites by S-phase-regulating kinases. *The Journal of biological chemistry* 281:10281–10290. doi:10.1074/JBC.M512921200

Rozpędek W, Pytel D, Nowak-Zduńczyk A, Lewko D, Wojtczak R, Diehl JA, Majsterek I. 2019. Breaking the DNA Damage Response via Serine/Threonine Kinase Inhibitors to Improve Cancer Treatment. *Current medicinal chemistry* 26:1425–1445. doi:10.2174/0929867325666180117102233

Thakur BL, Baris AM, Fu H, Redon CE, Pongor LS, Mosavarpour S, Gross JM, Jang S-M, Sebastian R, Utani K, Jenkins LM, Indig FE, Aladjem MI. 2022. Convergence of SIRT1 and ATR signaling to modulate replication origin dormancy. *Nucleic acids research* 50:5111–5128. doi:10.1093/NAR/GKAC299